# Perioperative Myocardial Infarction/Myocardial Injury Is Associated with High Hospital Mortality in Elderly Patients Undergoing Hip Fracture Surgery

**DOI:** 10.3390/jcm9124043

**Published:** 2020-12-14

**Authors:** Carlo Rostagno, Alessandro Cartei, Gaia Rubbieri, Alice Ceccofiglio, Agnese Magni, Silvia Forni, Roberto Civinini, Alberto Boccaccini

**Affiliations:** 1SODc Medicina Interna e Post-Chirurgica, AOU Careggi, 50136 Firenze, Italy; carteia@aou-careggi.toscana.it (A.C.); rubbierig@aou-careggi.toscana.it (G.R.); ceccofiglioa@aou-careggi.toscana.it (A.C.); agnese.magni@unifi.it (A.M.); 2Agenzia Regionale Sanità, 50136 Firenze, Italy; silvia.forni@ars.toscana.it; 3SOD Ortopedia e Traumatologia AOU Careggi, 50136 Firenze, Italy; roberto.civinini@unifi.it; 4SOD Anestesia e Rianimazione AOU Careggi, 50135 Firenze, Italy; boccaccinia@aou-careggi.toscana.it

**Keywords:** myocardial ischemia, troponin, hip fracture

## Abstract

Cardiovascular complications in patients undergoing non-cardiac surgery are associated with longer hospital stays and higher in-hospital mortality. The aim of this study was to assess the incidence of in-hospital myocardial infarction and/or myocardial injury in patients undergoing hip fracture surgery and their association with mortality. Moreover, we evaluated the prognostic value of troponin increase stratified on the basis of peak troponin value. The electronic records of 1970 consecutive hip fracture patients were reviewed. Patients <70 years, those with myocardial infarction <30 days, and those with sepsis or active cancer were excluded from the study. Troponin and ECG were obtained at admission and then at 12, 24, and 48 h after surgery. Echocardiography was made before and within 48 h after surgery. Myocardial injury was defined by peak troponin I levels > 99th percentile. A total of 1854 patients were included. An elevated troponin concentration was observed in 754 (40.7%) patients in the study population. Evidence of myocardial ischemia, fulfilling diagnosis of myocardial infarction, was found in 433 (57%). ECG and echo abnormalities were more frequent in patients with higher troponin values; however, mortality did not differ between patients with and without evidence of ischemia. Peak troponin was between 0.1 and 1 µg/L in 593 (30.3%). A total of 191 (10%) had peak troponin I ≥ 1 µg/L, and 98 died in hospital (5%). Mortality was significantly higher in both groups with troponin increase (HR = 1.37, 95% CI 1.1–1.7, *p* < 0.001 for peak troponin I between 0.1 and 1 µg/L; HR = 2.28, 95% CI 1.72–3.02, *p* < 0.0001 for peak troponin ≥1 µg/L) in comparison to patients without myocardial injury. Male gender, history of coronary heart disease, heart failure, and chronic kidney disease were also associated with in-hospital mortality. Myocardial injury/infarction is associated with increased mortality after hip fracture surgery. Elevated troponin values, but not ischemic changes, are related to early worse outcome.

## 1. Introduction

The incidence of perioperative myocardial infarction (MI) in patients with non-cardiac surgery has been reported between 6% and 36% according to the diagnostic criteria used in different studies [1,2,3]. Definite diagnosis of MI according to fourth universal definition requires, other than rise and fall of troponin values, at least one of the following conditions: typical symptoms, new electrocardiographic changes suggesting ischemia, or new segmental left ventricular wall abnormalities [4]. Most patients with perioperative MI/myocardial injury are asymptomatic. Moreover, after non-cardiac surgery ECG monitoring and/or echocardiography are not routinely performed. Evidence of “myocardial injury” mainly relies on post-operative increase of troponin values [5], and the diagnosis of true MI is underestimated.

Even small troponin increases (above upper 99th percentile of normal reference population) are associated with prolonged hospitalization and increased short- and long-term mortality [6,7]. Several clinical conditions, such as pulmonary embolism, sepsis, renal failure, and acute respiratory failure, may cause troponin to increase; however, these conditions may be ruled out on clinical basis. Patients undergoing hip surgery have a high incidence of myocardial injury, and troponin increase is associated with poor prognosis [8,9,10]. In our institution, patients undergoing hip surgery are followed by a multidisciplinary group [11]. This organization allows us to routinely perform post-operative ECG and echocardiogram; therefore, troponin changes can be related with clinical and/or objective evidence of myocardial ischemia, allowing for a distinction to be made between MI and myocardial injury. The aim of the present study was to assess the incidence of perioperative MI or myocardial injury and their relative effects on in-hospital mortality in patients who underwent hip fracture surgery. Furthermore, we evaluated the relation between different degrees of peak troponin values and early outcome.

## 2. Methods

This study is part of a project of the Italian Health Ministry and Regione Toscana—RF-2010-2316600—and was approved by the Ethical Committee of Regione Toscana. At admission, all patients gave signed informed consent in order to collect and analyze clinical data for research purposes. The study was conducted according to STROBE statements.

The primary endpoint of the study was to assess the incidence of perioperative MI and myocardial injury in patients undergoing hip fracture surgery and their relation to in-hospital mortality. A secondary endpoint was to evaluate the prognostic value of troponin increase stratified on the basis of peak troponin value. Finally, we evaluated the association of different clinical variables with early mortality in patients with perioperative MI/myocardial injury.

All patients with hip fracture referred to AOU Careggi Firenze within a five-year period were eligible for the study. At hospital admission, they underwent multidisciplinary clinical evaluation, as previously reported [11]. ECG and laboratory examination including troponin I assay were performed at admission. Siemens Dimension Vista^®^ System Flex^®^ reagent (Siemens Healthcare GmbH, Erlangen, Germany) was used for TnI assay. Troponin and ECG were repeated at 12, 24, and 48 h after surgery. Echocardiography was performed on all patients before surgery, and within 48 h thereafter in patients with troponin increase. Myocardial damage was defined by troponin I level above 99th percentile of normal distribution (>0.1 μg/L). Definite myocardial infarction was diagnosed on site by the two cardiologists of the group according the “fourth universal definition of myocardial infarction” [4]. Results were compared with patients with no myocardial injury (troponin I peak below 0.1 μg/L). Exclusion criteria were age <70 years, recent myocardial infarction (<30 days), sepsis, and active neoplastic disease. Finally, patients with diagnosis of takotsubo syndrome confirmed by absence of critical coronary stenosis at angiography were not included in the study. In the other patients, the increase in troponin, being an expression of an imbalance between myocardial oxygen demand and supply, was considered of ischemic origin.

Patients with troponin increase were stratified in two groups: the first with peak troponin between 0.1 and 1 μg/L, and the second with peak troponin above 1 μg/L. A cut-off value of 1 μg/L was used, since it was demonstrated to be a strong independent predictor of mortality [12].

Electronic records of patients were reviewed. The data set included demographic data, time to surgery, comorbidities—including atrial fibrillation, hypertension, diabetes mellitus, chronic renal failure (e.g., creatinine clearance <30 mL/min), CAD, heart failure, oral anticoagulant therapy (e.g., assumption of antivitamin K or direct oral anticoagulant agents like dabigatran, apixaban, rivaroxaban, or edoxaban at the moment of trauma), bleeding, sepsis, history of neoplastic disease, and cognitive impairment—troponin values, and ECG and echocardiographic findings. There were no missing data.

### Statistical Analysis

Categorical variables are reported as frequency and percentage. Values for continuous variables were given as the mean (+SD). Categorical variables were compared using chi-square or Fisher exact tests, while continuous variables were compared using Student’s *t*-test. The relation of multiple demographic and clinical variables with clinical outcome in patients with troponin increase was assessed by logistic multivariate analysis performed using SPSS 18.0 (IBM, Armonk, NY, USA) or Stata 14.1 (Stata Corp, College Station, TX, USA) statistical software. A probability value of <0.05 was considered to be statistically significant.

## 3. Results

### 3.1. Descriptive Results

A total of 1970 patients with hip fracture were referred to the Trauma Centre of Careggi University Hospital between 1 January 2014 and 31 December 2018 (30% male and 70% female, mean age 84 years). According to criteria reported in Section 2, 108 were excluded from the study (Figure 1). Five patients had sepsis, 48 active cancer. None was diagnosed with acute pulmonary embolism, while the 86 patients who developed atrial fibrillation after surgery were included, since we cannot exclude a relation between the onset of arrhythmia and transient perioperative ischemia.

Seven hundred fifty-four out of 1854 patients (40%) had a peak troponin value above the upper 99th percentile of the reference population (TnI > 0.1 µg/L), while 593 (30%) had peak troponin values between 0.1 and 1 µg/L (mean age 84.67 ± 6.1 years). Troponin peak I ≥ 1 µg/L was found in 191 (10%) patients (mean age 83.66 ± 6.4 years). Only five patients had typical chest pain; more common, but not specific signs, were hypotension and tachycardia.

History of coronary heart disease, heart failure, atrial fibrillation other than ongoing oral anticoagulation (antivitamin K or direct oral anticoagulants—DOACs), and antiplatelet treatment were significantly more frequent in patients with troponin >1 µg/L in comparison to the other two groups. Moreover, in this group the relative prevalence of males was significantly higher (82–43%, *p* < 0.001) than in controls and in patients with lower troponin values. Incidence of major bleeding (bleeding requiring red blood cell transfusion) was more frequent in patients with troponin changes than in those with no myocardial injury. The three groups did not show differences regarding the incidence of chronic kidney disease, history of cancer, or presence of cognitive impairment (Table 1).

ST-T changes suggesting myocardial ischemia (evidence of new onset ST segment depression, or of inverted or biphasic T wave) were found in less than 40% of patients with troponin values between 0.1 and 1 µg/L, and in 71% of patients with troponin peak >1 µg/L. The increase in troponin values was not associated with diagnostic changes at post-operative ECG in 65% of patients (Table 2).

Similarly, new left ventricular wall motion abnormalities were found more frequently in patients with peak troponin 1 µg/L (61% vs. 38% in patients with troponin values between 0.1 and 1 µg/L). Left ventricular ejection fraction was not significantly different in the two groups. In both groups, the incidence of severe aortic stenosis (valve area <1 cm^2^) was not negligible (13% and 16%, respectively).

Definite diagnosis of MI, according to fourth universal definition, was made in 433 (23%) patients after hip fracture surgery, while in 321 (17%) troponin changes were not associated with evidence of ischemia (myocardial injury). Diagnosis of MI was significantly more frequent in patients with troponin >1 µg/L (72 vs. 40%, *p* < 0.001).

Thirty patients underwent early coronary angiography, on average within five days after hip surgery. In 29, we found severe coronary artery disease (critical stenosis of at least two coronary vessels). The extent of coronary involvement was not related with the degree of troponin increase, although more frequently it was performed in patients with peak troponin ≥1 µg/L.

### 3.2. In-Hospital Outcome

Overall mortality during hospitalization was 98/1854 patients (5%). Mortality was significantly higher in both of the groups with increased troponin values in comparison to the control group. In fact, 39 patients (7%) with troponin I values between 0.1 and 1 µg/L and 26 (14%) with troponin I ≥ 1 µg/L died in hospital, in comparison to 32 (2.9%) controls (*p* < 0.001). Death was due to cardiovascular causes (mainly refractory heart failure and sudden death) in 75% of cases, and occurred to 82% of patients with troponin increase in comparison to 40% of patients with no myocardial injury.

We did not find significant differences in mortality between patients with electrocardiographic and/or echocardiographic ischemic changes (patients with definite diagnosis of “myocardial infarction”) or without them (patients with diagnosis of myocardial “injury”) (*p* = 0.364). When groups were stratified according to peak troponin values, survival was lower in patients with myocardial injury and troponin peak between 0.1 and 1 µg/L (Table 3).

Although hip fracture is largely prevalent in females (about 70% in our study), the relative proportion of male patients was significantly higher in those with increased troponin I (43%).

Table 4 indicates the main differences in clinical variables between patients discharged alive and those who died in hospital.

Serum creatinine did not significantly change after surgery. The absolute decrease in hemoglobin concentration was not significantly different among groups; however, bleeding requiring transfusion was more frequent in patients who went on to die. Although these patients were more frequently under oral anticoagulation, before the intervention INR was below 1.3 in all patients being treated with warfarin, while in patients treated with DOACs we followed the current guidelines regarding the time from last administration to surgery [13,14].

Logistic multivariate analysis showed that male gender, peak troponin values, history of coronary heart disease, heart failure, atrial fibrillation, and chronic kidney disease were associated with in-hospital mortality in patients undergoing hip fracture surgery (Table 5).

## 4. Discussion

### 4.1. Main Evidence and Comparison with Literature

Results from the present investigation confirm the association of perioperative troponin peak with an increased mortality in patients who underwent hip fracture surgery. Nevertheless, we did not find significant difference in mortality between patients with definite MI (diagnosed according to the fourth universal definition of myocardial infarction) and patients with isolated myocardial injury (patients with troponin increase without ECG and/or echocardiographic changes). Male gender, history of coronary heart disease, heart failure, atrial fibrillation, and chronic kidney disease were associated with an increased risk of early mortality in patients with myocardial infarction/myocardial damage.

Perioperative MI significantly affects length of hospitalization and hospital mortality in patients undergoing non-cardiac surgery [1,2,3,14]. Troponin measurement is currently recommended in high-risk patients after non-cardiac surgery [15,16,17,18]. Several investigations demonstrated that troponin increase is independently related to early and long-term mortality [19,20,21]. However, in absence of electrocardiographic monitoring and/or echocardiography, troponin increase was also frequently related to several causes other than ischemic myocardial injury (mainly pulmonary embolism, sepsis, renal failure, and acute respiratory failure).

Elderly patients undergoing surgery for hip fracture are at particular risk due to aging and comorbidities. Surgical treatment of hip fracture within 48 h from trauma is associated with higher probability of functional recovery and decrease of complication rate and overall mortality [22,23].

After surgery for hip fracture, the incidence of perioperative MI/myocardial injury ranges from 6% to 36% according to diagnostic criteria adopted.

In a retrospective study from Olmsted County, Minnesota [24], incidence of perioperative MI was 14%. Most patients did not experience ischemic symptoms, and required cardiac biomarkers for the diagnosis. In-hospital mortality was 14%. In the study by Hietala et al. [9], TnT increased in 35.5% of patients with hip fracture. Overall, 30-day mortality was 17% and increased to 24% in subgroup with values >0.15 μg/L. Troponin increase was the only independent predictor of 30-day mortality. Similarly, in the study by Fisher et al. [12], perioperative increase of troponin I > 0.06 μg/L proved to be an independent factor in length of hospital stay and need for long-term residential care facilities. Troponin I > 1 μg/L was a predictor of all-cause mortality with 98% specificity and 89% negative predictive value.

### 4.2. Diagnosis and Management

Multidisciplinary management may allow to identify preoperatively high-risk patients and favor clinical stabilization, thus improving safety of early surgery. In the postoperative period, the aim of this model is to limit, identify, and treat clinical complications. A multidisciplinary model has been employed in our teaching hospital since 2011. We conceived several clinical protocols, including one for diagnosis of perioperative MI/myocardial injury. Implementation of this multidisciplinary team was associated with a significant decrease in perioperative mortality [25].

The main novelty of the present investigation in comparison to previous studies was that our model, in which patients were followed by internal medicine specialists/cardiologists in the perioperative period, allowed for a more accurate differential diagnosis between definite MI and myocardial injury.

Overall incidence of MI according to the fourth universal definition was 19%. Echocardiography allowed us to make a diagnosis of MI in about 30% of subjects with troponin increase but with non-diagnostic changes at ECG.

Mortality was not significantly different between patients with and without definite diagnosis of MI. Otherwise, mortality was significantly different when patients were stratified according to peak troponin values. Troponin values ≥ 1µg/L were associated with a two-fold higher mortality in comparison to patients with increase of a lower degree.

Identification of high-risk patients based on clinical history and preoperative evaluation may be helpful in the choice of surgical and anesthesia strategies to limit incidence of perioperative MI and related mortality. Medical treatment, aspirin, statin, and, recently, dabigatran [26,27,28] have been demonstrated to decrease mortality related to perioperative MI in non-cardiac surgery. Aspirin and statin decreased the risk for 30-day mortality among patients who had suffered a perioperative MI (adjusted odds ratios 0.54, 95% CI 0.29–0.99 and 0.26, 95% CI 0.13–0.54, respectively) [26]. The design and results of the MANAGE study [26] do not allow definite conclusions about the usefulness of dabigatran in patients with postoperative troponin elevation after non-cardiac surgery (late enrollment, 35 days after surgery, low-risk profile, mean age 45 years, 14% history of coronary artery disease). The present study was not designed to evaluate medical interventions in order to improve the outcome in patients undergoing hip fracture surgery. Guidelines were followed (beta-blockers and antiplatelet drugs were not withdrawn before surgery in high-risk patients, and statin and ACE-I or ARB were maintained in patients with controlled/high blood pressure).

Diagnostic work-out in MI /myocardial injury after non-cardiac surgery is unusual, and the few studies have reported discordant data about the usefulness of early coronary angiography and revascularization [29,30]. The study of Parashar et al. [29] showed no benefit of early coronary revascularization in patients with postoperative MI after non-cardiac surgery. The outcome was poorer in particular in patients with major bleeding, chronic renal damage, and peripheral vascular disease. It must be emphasized that most patients had vascular, thoracic, and abdominal surgery. In our cohort, 30 patients underwent early coronary angiography. In 29 we found at least two coronary vessel diseases. We did not find a relation between degree of troponin increase and extension of coronary disease. This suggests that most patients with troponin increase have type II MI (myocardial injury) and severe atherosclerosis as the main pathogenetic mechanism of troponin release.

In a preliminary study from our group involving a small number of patients after hip surgery, early revascularization after perioperative myocardial infarction was associated with significant increase in one-year survival [28]. Larger studies are needed to assess the effects on early, in-hospital outcome.

## 5. Limitations

The main limitation of the present investigation was its retrospective design. Moreover, it was a single-center study including a relatively small number of patients and with a low number of events, although it must be underlined that most previous investigations of this topic had similar numerosity. Finally, troponin I concentrations were measured since hs-TnT has been available in our hospital only in the second half of 2019. At present, no studies directly compared standard TnI–TnT assay with hs-Tn in non-cardiac surgery. The higher sensitivity demonstrated in acute coronary syndromes must still be demonstrated in this setting. An investigation protocol has recently been submitted to the Ethical Committee of our hospital.

Nevertheless, a merit of this present study is that a definite protocol for preoperative evaluation of hip fracture patients and for diagnosis of perioperative MI/myocardial injury has been followed in our institution since 2014. This allowed a more accurate distinction between definite MI and myocardial injury. Due to organizational problems, extensive cardiologic follow-up with these patients has not been possible, and was limited to small subgroups. Therefore, in the whole population we were only able to analyze in-hospital outcome. Results from the present investigation may offer elements for further research on prevention of cardiac damage related to non-cardiac surgery. The finding of severe coronary artery disease in almost all patients who underwent coronary angiography may suggest coronary atherosclerosis as the main pathogenetic factor leading to troponin increase, and procedural interventions aimed to decrease the imbalance between oxygen demand and supply, as well as to plaque stabilization, may be hypothesized to decrease early mortality. We must agree that the relevance of these data is limited by the small number of patients who underwent coronary angiography.

## 6. Conclusions

Incidence of perioperative MI/myocardial injury after hip surgery was about 40%, and a definite diagnosis of myocardial infarction according to the fourth universal definition was made in 23% of patients. Higher peak troponin I level is related to a worse outcome. In fact, in-hospital mortality was as high as 14% in patients with troponin I > 1 μg/L, in comparison to 7% of those with increase of a lower degree. Otherwise, despite a tight protocol, we did not find significant differences in the outcomes between patients with definite MI and patients with myocardial injury. The diagnosis of MI might be slightly underestimated, however, since transient ECG changes may have been overlooked.

Future research should assess whether optimization of preoperative medical treatment, choice of anesthesia strategy, and decrease in intraoperative blood loss may limit incidence of perioperative MI/myocardial injury and related in-hospital deaths.

## Figures and Tables

**Figure 1 jcm-09-04043-f001:**
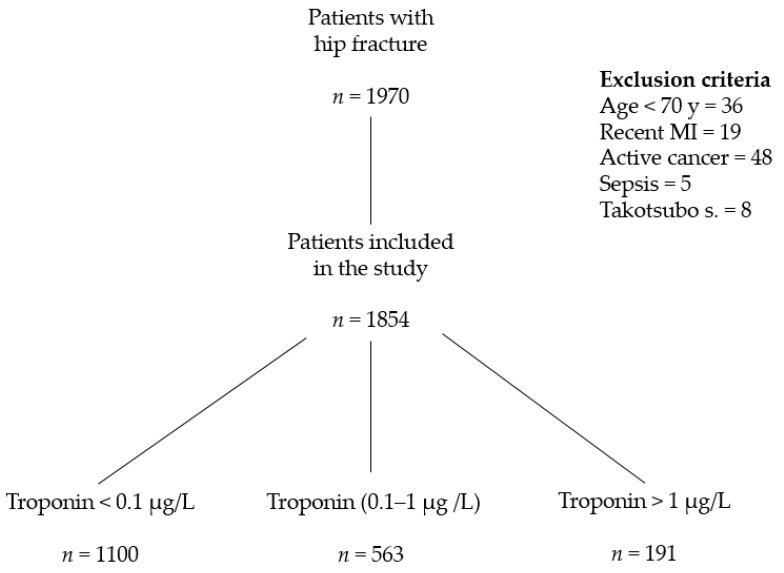
Patients included in the study.

**Table 1 jcm-09-04043-t001:** Demographic and clinical characteristics of the three groups under study.

Clinical Characteristics	No Myocardial Injury *n* = 1100	TnI 0.1–1 µg/L *n* = 563	TnI ≥ 1 µg/L *n* = 191	*p*-Value
Mean age (SD)	84.4 ± 6.1	84.67 ± 7.1	83.66 ± 6.4	0.527
Male	323 (29%)	167 (29%)	84 (43%)	0.0004
Atrial fibrillation	142 (12.9%)	109 (16%)	52 (28%)	<0.0001
Creatinine clearance <30 mL/min	162 (14%)	86 (15%)	41 (21%)	0.0506
Coronary disease	131 (12%)	121 (21%)	93 (48%)	<0.0001
Heart failure	92 (8%)	71 (12%)	37 (19%)	<0.0001
Oral anticoagulant treatment	116 (10%)	82 (14%)	40 (22%)	<0.0001
History of cancer	153 (14%)	106 (18%)	34 (17%)	0.1240
Cognitive impairment	265 (24%)	126 (23%)	52 (27%)	0.3795

SD = standard deviation.

**Table 2 jcm-09-04043-t002:** ECG and echocardiographic abnormalities in relation to degree of troponin increase.

ECG Changes	TnI 0.1–1 µg/L *n* = 563	TnI ≥ 1 µg/L *n* = 191	*p*-Value
Electrocardiogram			
Not diagnostic	337 (60%)	54 (28%)	<0.0001
ST-T changes *	219 (39%)	135 (71%)	<0.0001
ST elevation	6 (1%)	2 (1%)	0.9
Echocardiogram			
Not diagnostic	276 (49%)	44 (23%)	<0.0001
LV wall motion Abnormalities	214 (38%)	116 (61%)	<0.0001
Severe aortic stenosis	73 (13%)	31 (16%)	0.2749
Left ventricular EF (SD)	51 ± 6.2	50.3 ± 7	0.456

* evidence of new onset ST segment depression, or of inverted or biphasic T wave. LV = left ventricle; EF = ejection fraction.

**Table 3 jcm-09-04043-t003:** In-hospital mortality (%) per MI/myocardial injury (%) with electrocardiographic and/or echocardiographic ischemic changes and troponin level.

	MI/Myocardial Injury (%) with Electrocardiographic and/or Echocardiographic Ischemic Changes		*p*-Value
	Yes (433 Patients)	No (321 Patients)	
TnI 0.1–1 µg/L (563 patients)	13/290 (4.5%)	26/273 (9.5%)	0.0186
TnI ≥ 1 µg/L (191 patients)	21/143 (14%)	5/48 (10%)	0.4555

**Table 4 jcm-09-04043-t004:** Comparison of main comorbidities between patients with MI/MD who died in hospital or were discharged alive.

Clinical Characteristics	Patients Discharged Alive (*n* = 689)	Died (*n* = 65)	*p*-Value
Male gender	200 (29%)	41 (63%)	<0.0001
Time to surgery (<48 h)	510 (74%)	46 (70%)	0.383
Atrial fibrillation	179 (26%)	25 (38%)	0.02
Creatinine clearance <30 mL/min	103 (15%)	24 (36%)	<0.0001
Systolic pulmonary Pressure (>40 mmHg)	205 (29%)	20 (30%)	0.88
Coronary artery disease	187 (27%)	27 (41%)	0.005
Heart failure	83 (10%)	25 (38%)	<0.0001
Basic activities of daily living ≤4	265 (38%)	26 (40%)	0.79
Oral anticoagulation	85 (11%)	17 (26%)	0.002
Bleeding	165 (21%)	26 (40%)	<0.0001

**Table 5 jcm-09-04043-t005:** Factors related to early mortality at multivariate analysis.

Clinical Characteristics	HR	95% CI	*p*-Value
Age	1.15	0.94–1.42	0.07
Gender (female)	**0.57**	**0.47–0.68**	**<0.0001**
Time to surgery (<48 h)	0.95	0.89–1.05	0.398
Creatinine clearance <30 mL/min	**9.7**	**1.9–33**	**<0.0001**
Peak troponin 0.1–1 μg/L	**1.37**	**1.1–1.7**	**0.0005**
Peak troponin >1 μg/L	**2.28**	**1.72–3.02**	**<0.0001**
Systolic pulmonary artery pressure >40 mmHg	1.04	0.88–1.10	0.15
History of coronary artery disease	**5.3**	**1.03–25.4**	**0.04**
Atrial fibrillation	1.21	0.78–8	0.35
Basic Activities of Daily Living ≤4	0.98	0.68-1.41	0.935
Heart failure	**1.05**	**1.01–1.09**	**0.0081**
Bleeding	1.3	0.88–2.2	0.17
Oral anticoagulation	1.28	0.79–3.12	0.19

Bold: statistical significant factors related to early mortality

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
