# Peer review of "Perioperative Myocardial Infarction/Myocardial Injury Is Associated with High Hospital Mortality in Elderly Patients Undergoing Hip Fracture Surgery"

_jcm, 2020, doi:10.3390/jcm9124043_

Round 1

Reviewer 1 Report

I feel the edits in the manuscript have adequately addressed my previous comments.

Author Response

Dear Sir

I thank you for the help  in improving our paper 

Best regards on behalf of co authors

Carlo Rostagno 

Reviewer 2 Report

I would like to thank the authors for allowing me to review their manuscript. This study involves the recruitment of nearly 2000 patients over a 5 year period with hip fracture who had troponin testing and regular ECGs. It is a poignant paper in that there are large government-driven initiatives to reduce mortality in hip fracture, the outcome being a key indicator of institutional performance. Of note the study does have funding from the Italian Health Ministry.

It is a well written manuscript and flow logically. The authors find that a large proportion of patients have troponins elevated over the 99th centile, results consistent with other European studies performing ‘blanket’ testing. Previous cardiac disease, male sex and the prescription of antithrombotics are additionally associated with large troponin rises. The greater the troponin rise, the increased likelihood of developing both ECG changes indicative of AMI, and having echocardiogram abnormalities.

Inpatient mortality of 5% is similar to many other institutions, with a higher TnI rise correlated to higher mortality in univariate analysis. The authors outline that patients with a TnI increase over 1microgram/L had a similar inpatient mortality regardless of ECG or echo changes. Multivariate regression analysis of surgically managed patients showed that male sex, troponin rise, stage 4 kidney disease (or acute renal failure we assume) and a history of heart failure and coronary artery disease were independent predictors of inpatient mortality.

Overall, the paper has a message that many patients have TnI rises, and have worse outcomes. Treating teams should have a low threshold to examine for myocardial injury in hip fracture patients. None of the independent variables in the regression analysis are reversible.  The paper does not offer solutions to this, however, does cite potentially cardioprotective therapies such as antithrombotics, betablockers and statins. Despite referencing many non-cardiac surgery studies, the effectiveness of such therapies in hip fracture is not known and should be the focus of future studies.

Suggestions below

“we followed the current guideline regarding the time from last administration to surgery”

Please reference these guidelines in the methods. Timing of surgery for DOACs is a point of contention, and there is a multitude of guidelines based on little evidence.

“Figures and Tables.”

Please include all abbreviations within a table in the table legend below.

For Table 5, can you please put the HR and 95% CI in the first two columns before the p values.

There are some minor formatting inconsistencies, such as “TnI 0,1-1μg/L” in one table, then “TnI 0.1-1μg/L” in the next. Please try to stick to the English rather than European formatting for what it is worth.

Figure 2 probably doesn’t add a great deal, and if the information can be incorporated in the text, it would be preferable.

“Formatting”

Stick to a format for numbers. Some values have decimal places, others don’t. I would personally keep all  numbers below 10 to 1 decimal place, and all numbers above 10 to no decimal places. Eg. 21/143 (14.7%) in Table 3 is different to 73 (13%) in Table 2. It’s a simple revision that makes the documents flow better.

For p values, please keep to 3 significant figures. In the regression analysis in table 5, a p values of .00001 is not needed. <0.001 is enough accuracy for the reader to understand that being male will lead to a bad outcome.

“Nevertheless, we did not find significant difference in mortality between patients with definite MI, diagnosed according to 4th universal definition of myocardial infarction, and patients with isolated myocardial injury (patients with troponin increase without ECG and/or echocardiographic changes).”

Did you not find a difference for those with a TnI of 0.1-1 microgram/L? There was just no difference with a value of 1 mcg/L and above. Suggesting that further testing won’t change the outcome with such severely damaged myocardium, whether there is evidence of coronary thrombosis or not.

“non cardiac surgery [1-3,13]. Troponin measurement is currently recommended in high risk patients after non-cardiac surgery”
Make sure there is the same formatting for hyphenated terms.

“Implementation of this multidisciplinary team was associated with a significant decrease of perioperative mortality [24].”

I believe you should be referring to reference 23.

The discussion is otherwise reasonable. It may benefit from subheadings to give it direction and address each of the aims. 

“Conclusions”

Please make this shorter. I would only address the feature of your hypothesis. I would include one in the introduction personally, but in the absence of one, at least answer your aim. This was to “assess the incidence of perioperative MI or myocardial injury and their relative effects on in-hospital mortality in patients underwent hip fracture surgery. Furthermore, we evaluated the relation between different degrees of peak troponin values and early outcome.”

Author Response

“we followed the current guideline regarding the time from last administration to surgery”

Please reference these guidelines in the methods. Timing of surgery for DOACs is a point of contention, and there is a multitude of guidelines based on little evidence.

In the reference section we reported EHRA guidelines and suggestions form the paper of Douketis et al

Steffel J, Verhamme P, Potpara TS, et al. The 2018 European Heart Rhythm Association Practical Guide on the use of non-vitamin K antagonist oral anticoagulants in patients with atrial fibrillation. Eur Heart J. 2018;39(16):1330‐1393.

Douketis JD, Spyropoulos AC, Duncan J, et al. Perioperative Management of Patients With Atrial Fibrillation Receiving a Direct Oral Anticoagulant [published online ahead of print, 2019 Aug 5]. JAMA Intern Med. 2019;179(11):1469‐1478.

“Figures and Tables.”

Please include all abbreviations within a table in the table legend below.

Abbreviations were reported in table legends

For Table 5, can you please put the HR and 95% CI in the first two columns before the p values.

The change was made according to your suggestions

There are some minor formatting inconsistencies, such as “TnI 0,1-1μg/L” in one table, then “TnI 0.1-1μg/L” in the next. Please try to stick to the English rather than European formatting for what it is worth.

The paper was checked and formatting uniformed to English

Figure 2 probably doesn’t add a great deal, and if the information can be incorporated in the text, it would be preferable.

 The figure was removed, the information are incorporated in the text

“Formatting”

Stick to a format for numbers. Some values have decimal places, others don’t. I would personally keep all  numbers below 10 to 1 decimal place, and all numbers above 10 to no decimal places. Eg. 21/143 (14.7%) in Table 3 is different to 73 (13%) in Table 2. It’s a simple revision that makes the documents flow better.

 The changes were made according to your suggestions

For p values, please keep to 3 significant figures. In the regression analysis in table 5, a p values of .00001 is not needed. <0.001 is enough accuracy for the reader to understand that being male will lead to a bad outcome.

 The changes were made according to your suggestions

“Nevertheless, we did not find significant difference in mortality between patients with definite MI, diagnosed according to 4th universal definition of myocardial infarction, and patients with isolated myocardial injury (patients with troponin increase without ECG and/or echocardiographic changes).”

Did you not find a difference for those with a TnI of 0.1-1 microgram/L? There was just no difference with a value of 1 mcg/L and above. Suggesting that further testing won’t change the outcome with such severely damaged myocardium, whether there is evidence of coronary thrombosis or not.

When we look to the whole population of patients with increased troponin we did not find prognostic difference between the patients with or without evidence of myocardial ischemia and this is the message we gave in discussion.  Effectively  patients with a TnI of 0.1-1 microgram/L without ECG and ECHO abnormalities had an higher mortality  than those with evidence of ischemia and this was a surprise .  The limited number of events  however did  not allow to draw conclusions  about a causal mechamism.

“non cardiac surgery [1-3,13]. Troponin measurement is currently recommended in high risk patients after non-cardiac surgery”
Make sure there is the same formatting for hyphenated terms.

  The changes were made according to your suggestions

“Implementation of this multidisciplinary team was associated with a significant decrease of perioperative mortality [24].”

I believe you should be referring to reference 23.

This is correct and the reference now matches the text

The discussion is otherwise reasonable. It may benefit from subheadings to give it direction and address each of the aims. 

Two subheadings were added to discussion

“Conclusions”

Please make this shorter. I would only address the feature of your hypothesis. I would include one in the introduction personally, but in the absence of one, at least answer your aim. This was to “assess the incidence of perioperative MI or myocardial injury and their relative effects on in-hospital mortality in patients underwent hip fracture surgery. Furthermore, we evaluated the relation between different degrees of peak troponin values and early outcome.”

Conclusions were modified according to your suggestions

This manuscript is a resubmission of an earlier submission. The following is a list of the peer review reports and author responses from that submission.

Round 1

Reviewer 1 Report

I believe the manuscript has benefitted from the revisions and merits publication in the journal.

Author Response

Thank you for your appreciation

Reviewer 2 Report

Thank you for the opportunity to review the manuscript by Rostagno et al. regarding the incidence of myocardial damage and/or MI and its relation to mortality after hip fracture surgery. This subject has to date already been addressed in several studies, however, in this study by Rostagno et al. a systematic echocardiography was performed before and after surgery – this adds some new data to the area.

However, I have several major and minor concerns.

In general, the language and grammar should be further improved. Several sentences are complicated to read and several prepositions are lacking.

The structure should adhere to the Strobe guidelines

The aim of the study is still not clear. Please rephrase. A suggestion: The aim of the study was to assess the incidence of in-hospital myocardial infarction and/or myocardial damage in patients undergoing hip fracture surgery and the association between peak troponin, ischemic ecg changes or echocardiographic signs of myocardial ischemia and mortality.

Abstract

Several key points are missing:

Add the aim of the study

Specify the primary and secondary outcomes clearly e.g. the primary outcome of the study was….

Add a p-value to the comparison of the incidence of mortality

Add numbers and/or percentages to the results e.g. 754 (40.7%) had troponin increase

Methods

Please change “Experimental section” to “Methods”

It is not clear whether the study only included patients with a hip fracture

Was the primary outcome myocardial damage due to myocardial ischemia? As the authors state, several non-ischemic causes can cause troponin leak. Could the authors clarify the primary outcome.

Which troponin assay was used?

Why was 1 μg/L chosen as the troponin cut-off between the two troponin groups? References?

Statistics: How did the authors check for normality? Was all continuous variables normally distributed since median (iqr) was not used. Why did the authors perform a univariate analysis? How does the analysis fit with the aim of the study? How did they chose the variables? Why was myocardial damage and/or MI not included as a variable?

Results

Normally baseline data (table 2) are presented in the first section of the results.

Add numbers and/or percentages – often only the number of patients or the percentages are written (please see comment in abstract)

Did any of the patients have subjective symptoms?

Please elaborate on the non-ischemic causes to troponin leak. How many patients has sepsis, pulmonary embolism, chronic troponin increase, an acute cardiac arrhythmia e.g. AF, acute kidney injury etc.

Table 2: what is ST-T changes? Please elaborate.

It could be interesting to see the echocardiographic findings in the control group (no troponin increase). Did any of the patients in the control group have changes? One could imagine that the surgical stress response in combination with preexisting cardiac comorbidities could induce a transient left ventricular dysfunction.

Page 5, line 11 “There was no relation with degree of troponin increase although coronary angiography was more frequently performed in patients with peak troponin ≥ 1 μg/L.” It is not clear what is meant.

Section 3.2. “Survival analysis”. You might consider to change the title of this section since you did not perform a survival analysis.

You mention that mortality was higher in patients with troponin elevations, however, I do not see any statistics beside crude percentages.

You mention bleeding. Do you have any biochemical data on the patients incl. s-creatinine and hemoglobin. This could be interesting.  

Discussion

A short resumé of your results should be included in the beginning of the discussion (please see Strobe guidelines)

The study is interesting due to the systematic echocardiography, however, this finding is not at all discussed. You found that 38% of the patients with minor troponin leaks have LV wall motion abnormalities and 61% of the patients with troponin ≥ 1 μg/. What implications do this finding have?

How could you explain that mortality did not differ between patients with myocardial damage and MI?

Conclusion

The conclusion should match the aim of your study. The relevant results are not mentioned.

Author Response

Abstract

Several key points are missing:

Add the aim of the study   The aim of the study is reported according to your suggestions

Specify the primary and secondary outcomes clearly e.g. the primary outcome of the study was….Outcomes are reported along with methods

Add a p-value to the comparison of the incidence of mortality.  P value was added

Add numbers and/or percentages to the results e.g. 754 (40.7%) had troponin increase. Numbers and percentages have been added.

Methods

Please change “Experimental section” to “Methods”. The change was made .

It is not clear whether the study only included patients with a hip fracture. The text was modified and now I hope that now may be clear that the study included only patients with hip fracture.

Was the primary outcome myocardial damage due to myocardial ischemia? As the authors state, several non-ischemic causes can cause troponin leak. Could the authors clarify the primary outcome.

The outcome of the study was clarified in the text.

Which troponin assay was used?  Siemens Dimension Vista® System Flex® reagent was used for TnI assay

Why was 1 μg/L chosen as the troponin cut-off between the two troponin groups? References?  As reported in the discussion in the study of Fisher et al (present reference n 12 )  a troponin value > 1 μg/L was associated with an high sensitivity and specificity in identifying patients at risk of death after hip fracture surgery.  This is also reported in methods section.

Statistics: How did the authors check for normality? Was all continuous variables normally distributed since median (iqr) was not used. Why did the authors perform a univariate analysis? How does the analysis fit with the aim of the study? How did they chose the variables? Why was myocardial damage and/or MI not included as a variable?

Continuous parameters included in the study are limited and they were normally distributed.In the previous version of the paper we performed also a multivariate analysis ( results are reported below) but the Acadeni Editor who reviewed the paper considered the analysis redundant and advised us to remove it from the paper

Table 5- Factors related to in-hospital mortality at multivariate analysis

Age

.07

1.15

0.94-1.42

Gender ( F vs M )

.00001

0.57

0.47-0-68

Time to surgery

.398

0.95

0.89-1.05

Creatinine clearance < 30 ml/min

.00007

9.7

1.9--33

Peak troponin 0.1-1 mg/L

.0005

1.37

1.1-1.7

Peak troponin > 1 mg/L

.00001

2.28

1.72-3.02

Pulmonary artery pressure

.15

1.04

0.88-1.10

History of coronary artery disease

.04

5.3

1.03-25.4

Atrial fibrillation

.35

1.21

0.78- 8

BADL

.935

0.98

0.68-1.41

Heart failure

.0081

1.05

1.01-1.09

Bleeding

.17

1.3

0.88-2.2

Oral anticoagulation

.19

1.28

0.79-3.12

Mayor comorbidities associated with hip fracture prognosis derived from our previous experience  were included in the analysis Now myocardial infarction/myocardial damage are reported in the table

Results

Normally baseline data (table 2) are presented in the first section of the results. The order of presentation wasn changed according to your suggestions

Add numbers and/or percentages – often only the number of patients or the percentages are written (please see comment in abstract) Number and percentages were added where needed

Did any of the patients have subjective symptoms? As already reported in literature symptoms of myocardial ischemia are uncommon. Only 5 patients had typical chest pain, more common were hypotension and tachycardia

Please elaborate on the non-ischemic causes to troponin leak. How many patients has sepsis, pulmonary embolism, chronic troponin increase, an acute cardiac arrhythmia e.g. AF, acute kidney injury etc. This was discussed 

Table 2: what is ST-T changes? Please elaborate. New ST depression, T inverted  or biphasic were considered as sign of myocardial ischemia . This was reported in the text and table

It could be interesting to see the echocardiographic findings in the control group (no troponin increase). Did any of the patients in the control group have changes? One could imagine that the surgical stress response in combination with preexisting cardiac comorbidities could induce a transient left ventricular dysfunction.

All patients underwent preoperative echocardiography to allow better preoperative risk stratification. However due to limited resources postoperative examination was performed routinely, for protocol,  only in patients with troponin increase.

Page 5, line 11 “There was no relation with degree of troponin increase although coronary angiography was more frequently performed in patients with peak troponin ≥ 1 μg/L.” It is not clear what is meant.

The sentence was reformulated and I hope that present version may be clear

Section 3.2. “Survival analysis”. You might consider to change the title of this section since you did not perform a survival analysis.

The title was changed.

You mention that mortality was higher in patients with troponin elevations, however, I do not see any statistics beside crude percentages.

The results were analyzed but we forgot to write  P value that was added

You mention bleeding. Do you have any biochemical data on the patients incl. s-creatinine and hemoglobin. This could be interesting.  

Serum creatinine did not significantly changed and non-significant changes in absolute decrease in hemoglobin concentration were found In the three groups.

Discussion

A short resumé of your results should be included in the beginning of the discussion (please see Strobe guidelines)

The synthesis of results was included in the beginning of discussion

The study is interesting due to the systematic echocardiography, however, this finding is not at all discussed. You found that 38% of the patients with minor troponin leaks have LV wall motion abnormalities and 61% of the patients with troponin ≥ 1 μg/. What implications do this finding have?

As previously reported by other authors also in different post-operative settings the prognostic value of troponin is proportional to the degree of the increase.  This may be associated with  a more extensive damage that may lead to clear changes of left ventricular wall motion.  The few data about coronary anatomy in these patients do not allow definite conclusions but we may hypothesize that the extent of coronary disease may be worse . Prospective study are needed to answer this question, 

How could you explain that mortality did not differ between patients with myocardial damage and MI?

We do not have a definite answer. The more likely explanation is that we missed transient ECG (or less probably echocardiographic ) changes and therefore incidence of myocardial infarction may be underestimated. However in the few cases of patients underwent coronary angiography in absence of definite diagnosis of myocardial infarction had similar extent of coronary lesions  .

Conclusion

The conclusion should match the aim of your study. The relevant results are not mentioned.

Conclusions were changed according to your suggestion,

Round 2

Reviewer 2 Report

The manuscript has improved, however, I still have some major concerns, which I do not believe that the authors have adressed in the previous revision. 

The language is still poor and extensive editing should be done. 

The aim is still not consistent throughout the manuscript (introduction vs. the abstract and methods)                                                                             

I suggested that the study should be reported according to the Strobe guidelines. The authors now added that study was conducted according to the strobe guidelines. I think the authors mean that the study was reported according to the strobe guidelines? However if so, important sections are missing from the strobe guidelines.

In my last revision, I asked for the primary and secondary outcomes to be clearly specified. In the methods section, it does not clearly say “the primary outcome was…. And the secondary outcomes were….”. Please add.

Please also clearly state in the methods whether the primary outcome, myocardial damage and/or inferction, was due to myocardial ischemia.

A p-value was added to the comparison of mortalities, however, not in the abstract.

Percentages are still missing in the results section

Please explain in the manuscript why and how the univariate analysis fits with the aim of your study. That an academic editor advised you to remove a multivariate analysis does not warrant a univariate analysis. What is the purpose? Please explain in the manuscript. For this reviewer it seems redundant if a multivariate analysis is not performed since the aim seems to be to study the potential association between myocardial damage/infarction and mortality in an adjusted model.

Moreover, please explain in the manuscript (statistics) how your model was constructed (why did you include the variables you did). This reviewer could argue that eg. a history of stroke should be included as well.

Two different tables are called “table 1” in the present version of the manuscript

Author Response

The language is still poor and extensive editing should be done.  Further editing was made, we hope that in present version it may be acceptable. 

The aim is still not consistent throughout the manuscript (introduction vs. the abstract and methods)    Aim was rewritten as required                                                                        

I suggested that the study should be reported according to the Strobe guidelines. The authors now added that study was conducted according to the strobe guidelines. I think the authors mean that the study was reported according to the strobe guidelines? However if so, important sections are missing from the strobe guidelines.  We checked point by point Strobe Guidelines and we now think that the paper may be adherent to Strobe statements,  

In my last revision, I asked for the primary and secondary outcomes to be clearly specified. In the methods section, it does not clearly say “the primary outcome was…. And the secondary outcomes were….”. Please add. Primary and secondary outcomes were added according to your suggestion.

Please also clearly state in the methods whether the primary outcome, myocardial damage and/or inferction, was due to myocardial ischemia.  Troponin leakage , whatever the favoring condition, is related to an imbalance between oxygen demand and supply, therefore  myocardial ischemia independent from  the degree of obstructive coronary disease .

A p-value was added to the comparison of mortalities, however, not in the abstract. The p-value was added in the abstract.

Percentages are still missing in the results section.  Percentages were added were missing

Please explain in the manuscript why and how the univariate analysis fits with the aim of your study. That an academic editor advised you to remove a multivariate analysis does not warrant a univariate analysis. What is the purpose? Please explain in the manuscript. For this reviewer it seems redundant if a multivariate analysis is not performed since the aim seems to be to study the potential association between myocardial damage/infarction and mortality in an adjusted model.  I admit that previous comments may had confounding results.  In present version considered variables were analyzed in survived and patients went to death in hospital and logistic multivariate analysis was restored to study their  potential association with  mortality in patients with myocardial damage/infarction

Moreover, please explain in the manuscript (statistics) how your model was constructed (why did you include the variables you did). This reviewer could argue that eg. a history of stroke should be included as well. The variables included in the study were chosen according to results of previous studies from our group Overall number of patients with previous stroke was below 4% and did not affect in-hospital outcome (graduation thesis Dr Michele Ciabatti) therefore it was not reported in present analysis. 

Two different tables are called “table 1” in the present version of the manuscript . I apologize for the mistake, the order was inverted in comparison to previous version

I hope you can appreciate the efforts made during a period of hard clinical work due to recrudescence of  SARS Cov 2 pandemia.